# Evaluating VLMs' General Capability on Next Location Prediction

## Abstract

Predicting the next location is a hallmark of spatial intelligence. In real-world scenarios, humans often rely on visual estimation to perform next-location prediction, such as anticipating movement to avoid collisions with others. With the emergence of large models demonstrating general visual capabilities, we explore whether vision-language models (VLMs) can perform similar next location prediction as human. We present **VLMLocPredictor**, a benchmark for evaluating VLMs on next location prediction tasks by contributing: (1) the Visual Guided Location Search (VGLS) module, a recursive refinement strategy leveraging visual guidance to iteratively narrow the search space for predictions; (2) a comprehensive vision-based dataset integrating open-source map taxi trajectory; (3) a human benchmark established via a large-scale social experiment. Through over 1000 queries on 14 VLMs, our findings indicate that VLMs exhibit promising potential for next-location prediction through our methods. However, their performance currently does not reach human-level accuracy. While some VLMs show potential to outperform humans in 24% scenarios, we believe in the near future, VLMs will surpass the average human performance in next-location prediction tasks. The benchmark and resources are available at `https://ihhh.cn`.

## 1. Introduction

Next-location prediction is a hallmark of spatial intelligence. Driven by environmental factors, temporal intervals, movement trends, and the intrinsic characteristics of moving entities, the next position of a trajectory exhibits a degree of predictability(Song et al., 2010). Interestingly, when

[1]Anonymous Institution, Anonymous City, Anonymous Region, Anonymous Country. Correspondence to: Anonymous Author <anon.email@domain.com>.

Preliminary work. Under review by the International Conference on Machine Learning (ICML). Do not distribute.

encountering scenes like anticipating movement to avoid collisions with others, humans often make these predictions not through mathematical calculations but through visual estimations. This observation suggests that next-location prediction is also a manifestation of visual intelligence.

With the rapid advancement of large models, vision-language models (VLMs) have demonstrated remarkable capabilities in comprehending and interpreting images(OpenAI, 2024a; Claude, 2024b; Gemini Team, 2024b). Some researchers even argue that these models exhibit early signs of artificial general intelligence (AGI)(Bubeck et al., 2023). This paper, therefore, investigates a foundational question: **Do vision-language models (VLMs) have the capability to make next-location predictions using vision intelligence like human?**

If VLMs possess similar capabilities, they also hold significant promise to the task of next location prediction. Traditionally, next-location prediction frameworks have relied on **training specialized models for specific cities using large volumes of data**. However, leveraging the general VLMs to perform universal next location prediction could enable the development of generalized models for next location prediction, transcending city-specific constraints.

To advance this investigation, we propose VLMLocPredictor, a new **vision-based** benchmark that focuses on next location prediction. This is a challenging task due to several factors: 1) The movement of objects is influenced by both their prior states and environmental constraints. 2) The trajectories of objects exhibit logical patterns. 3) Object motion is further constrained by temporal intervals. Although next-location prediction inherently involves uncertainty and cannot achieve 100% accuracy, human predictions often converge to a low error margin, demonstrating the spatial capability. The framework is shown in Figure 1.

In VLMLocPredictor, we first developed a Visual Guided Location Search (VGLS) module. This innovative module employs a recursive strategy of iterative binary partitioning of feasible regions. At each step, visual guidance is utilized to divide the current region into smaller subregions, prompting the VLM to assess and prioritize the subregion most likely to contain the next location, progressively narrowing the search space and enhancing prediction accuracy.

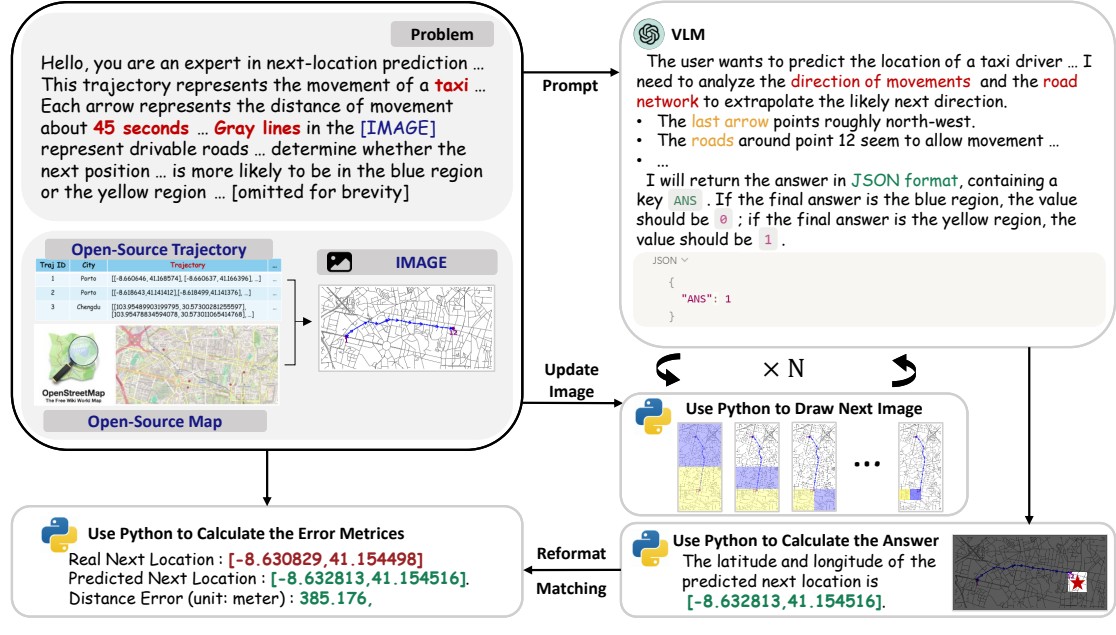

*Figure 1.* **Overview of VLMLocPredictor.** Using open-source trajectory and map datasets, we constructed trajectory images overlaid on road networks. For each trajectory, we designed visual guidance that is input into pre-prompted VLMs, followed by iterative questioning based on the model's responses. Through this multi-step dialogue with the VLM, we derived predictions for the next location point. These predictions were then compared with ground truth values to calculate the error.

Second, because taxi trajectories can be interpreted as moving objects, while road networks provide an environmental context for their operation, we constructed a comprehensive dataset by integrating open-source map data with publicly available taxi trajectory data. Using this dataset, we systematically evaluate the performance of 14 different VLMs. Each model's performance was assessed through meticulously curated 1160 queries.

Finally, we conducted a social experiment to evaluate human performance in the next location prediction. We developed an interactive trajectory prediction platform where participants were tasked with predicting the next point in the same test set as the large-scale models. This experiment has collected over 10000 results and provides a valuable benchmark for comparing human reasoning with model performance. The platform is publicly accessible at `https://ihhh.cn`.

Here are the main findings:

- Through the proposed method, VLMs are successfully endowed with the ability to predict the next location. The experimental results demonstrate that VLMs can produce meaningful predictions rather than random guesses. Moreover, they are capable of making complex decisions, such as taking turns, while also adhering to the constraints imposed by the road network.

- At present, the capabilities of VLMs still lag behind those of humans. However, the best-performing VLM has surpassed human performance in 24% scenarios and outperformed experts in 15% cases, which we consider an optimistic result.

- Based on the experimental results presented in this paper, we hypothesize that, in the near future, VLMs will approach the average human performance in next-location prediction tasks on urban maps, and may even surpass deep learning models and experts.

The goal of this study is to **evaluate the capabilities of general VLMs with training-free setting** and demonstrate that **VLMs possess the ability to understand maps and predict next locations**. Given that Large Language Models (LLMs) struggle to process complex map data(Wang et al., 2023a), we believe that our findings pave the way for the development of more effective next-location prediction models. We leave tuning specialized VLMs to be a promising direction for future work.

## 2. Preliminaries

Definition 1 (**Trajectory**): Let a trajectory $\mathbf{T}$ be defined as an ordered sequence of 13 spatial points, denoted as $\mathbf{T} = \{p_1, p_2, \ldots, p_{13}\}$. Each point $p_i$ ($i = 1, 2, \ldots, 13$) is

represented by its geographic coordinates $p_i = (lat_i, lon_i)$, where $lat_i$ and $lon_i$ correspond to the latitude and longitude values, respectively. To ensure consistency and applicability in next location prediction tasks, the trajectory is processed to be uniform with a constant temporal interval $\Delta t = 45s$.

Definition 2 (**Next Location Prediction**): Next Location Prediction is to learn a predictive function $f$ such that: $f(\mathbf{T}_{1:12}) = \hat{p}_{13}$, where $\hat{p}_{13}$ is the predicted location that approximates the ground-truth $p_{13}$.

## 3. VLMLocPredictor

In this section, we present the VLMLocPredictor, a comprehensive framework designed to evaluate the capabilities of VLMs in the Next Location Prediction task.

### 3.1. Visual Guided Location Search

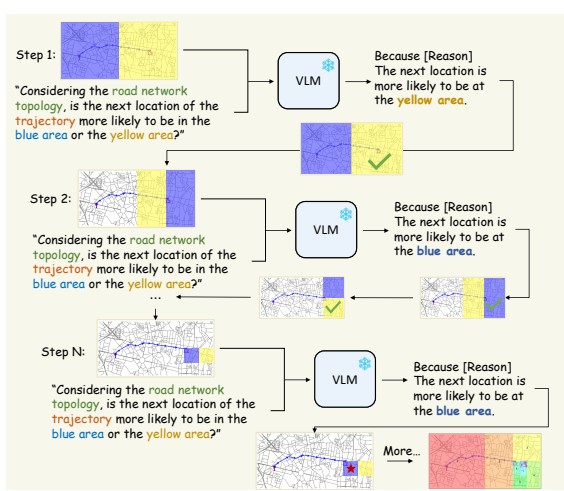

*Figure 2.* **An illustration of VGLS.** The image was initially divided into two halves, colored blue and yellow. After querying the VLM, the feedback was used to identify a probable region, which was then further divided. The process was iteratively repeated over multiple steps. Through this iterative feedback loop, the VLM eventually provided a feasible solution for the next location point.

To evaluate the capability of VLMs to predict the next location point, one critical challenge is how to equip these models with the ability to predict the next location. At the time of submission, VLMs do not inherently possess the ability to annotate or localize points on a map. **Given the goal of this study is to assess the visual intelligence of general VLMs, we deliberately avoid fine-tuning methods**, as such approaches would contradict the premise of evaluating general capabilities. Moreover, **the emphasis on visual intelligence precludes using explicit textual outputs**, since animals themselves cannot directly calculate the precise coordinates of the next point by texts.

We are inspired by related works that proved VLMs are capable of interpreting specific regions delineated by a circle(Shtedritski et al., 2023) and can handle tasks recursively(Wu & Xie, 2024). Building on these insights, we propose a Visual Guided Location Search (VGLS) mechanism. The core idea is as follows: we begin by dividing the map into two regions, coloring one half blue and the other half yellow. The VLM is then tasked with determining whether the next trajectory point is more likely located in the blue or yellow region. Take the model that identifies the blue region as more likely as an example, the blue region is further subdivided into one half colored blue and the other yellow. This iterative process is repeated for $N$ steps, progressively narrowing down the feasible region until a satisfactory resolution is achieved.

Fundamentally, we designed a hierarchical question-answering process. In the first iteration, the model only needs to answer a relatively simple question: which half of the map is more likely to contain the next point? As the process continues, the questions become increasingly challenging, with scenarios emerging where both blue and yellow regions may contain plausible solutions. This iterative refinement ensures a systematic localization process, leveraging the visual reasoning capabilities of VLMs.

### 3.2. Dataset

As the objective of this study is to evaluate VLMs by leveraging the next location prediction task, we choose to conduct our evaluation on taxi trajectory data for several key reasons: (1) Taxi trajectories are inherently constrained by the road network, as taxis primarily operate on predefined roads. (2) Taxi trips are typically goal-oriented, meaning that consecutive trajectory points exhibit strong logical dependencies. (3) There exist extensive open-source datasets of taxi trajectories and road networks, providing a rich resource for benchmarking VLMs in this task.

**Dataset Construction**: To this end, we selected two of the most commonly used taxi datasets: the Porto Taxi Dataset[1] and the DiDi (Chengdu) Dataset[2]. Both datasets comprise hundreds of thousands of taxi trajectories, providing a solid foundation for next location prediction. Additionally, we obtained the road network data for these two cities from OpenStreetMap[3]. Trajectories and road networks were overlaid on the images and underwent a series of post-processing and selection steps to ensure visibility.

**Dataset Separation**: Furthermore, through detailed data

---

[1] https://archive.ics.uci.edu/dataset/339/taxi+service+trajectory+prediction+challenge+ecml+pkdd+2015

[2] http://outreach.didichuxing.com/research/opendata/en/

[3] https://www.openstreetmap.org/

Table 1. The Statistic of Our Dataset.

|  | Easy | Medium | Hard |
|---|---|---|---|
| Number of Trajectories | 36 | 44 | 36 |
| Number of Queries | 360 | 440 | 360 |
| Average $D_{12,13}(m)$ | 186 | 298 | 510 |
| Average $D_{1,13}(m)$ | 1957 | 2818 | 3085 |
| Average Number of Roads | 690 | 1045 | 1639 |
| Average Square(km$^2$) | 6.94 | 9.3 | 10.08 |

analysis (see Appendix B.1), we identify a statistically positive correlation between the model's performance and two key factors: the number of roads present in the image ($N_{road}$) and the distance between the 12th and 13th trajectory points ($D_{12,13}$). Based on these findings, we partition the dataset into three subsets: **hard**, **medium**, and **easy**. Specifically, samples where both $N_{road}$ and $D_{12,13}$ exceed the median number are classified as hard, while those where both values fall below the median number are labeled as easy. The remaining samples constitute the medium category. The detailed statistics of these subsets are provided in Table 1. The table clearly demonstrates that the task difficulty levels we defined follow a progressive pattern,

**Prompt Consideration**: Given that the task involves feeding trajectory containing images into a VLM, prompt design plays a critical role. Since our goal is to evaluate the visual intelligence of VLMs, we deliberately avoid inputting GPS coordinates directly into the model. Including GPS coordinates might lead the model to bypass visual reasoning and solve the problem directly through textual input. Hence, we exclude explicit GPS information in the prompts to preserve the task's visual nature. The complete prompt can be found in the Appendix A.

### 3.3. Human Evaluation Website

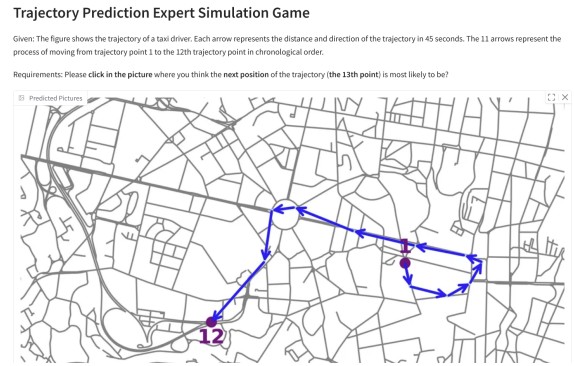

**Trajectory Prediction Expert Simulation Game**

Given: The figure shows the trajectory of a taxi driver. Each arrow represents the distance and direction of the trajectory in 45 seconds. The 11 arrows represent the process of moving from trajectory point 1 to the 12th trajectory point in chronological order.

Requirements: Please **click in the picture** where you think the **next position** of the trajectory (**the 13th point**) is most likely to be?

*Figure 3.* The screenshot of the website for human participants to predict the next location

To evaluate whether the capability of VLMs can rival human

intelligence, we conducted a large-scale human experiment. We developed an online platform where users are presented with the same input prompts and images as the VLMs. The task required users to directly select the most likely next location within the given image. This platform is shown in the Figure 3. By the time of this submission, the platform has gathered over 10,000 annotated next-location predictions from more than 100 participants. On average, each trajectory includes over 100 human predictions, and the data collection is still ongoing.

### 3.4. Evaluation

To compare our method with various baselines, we first select metrics that enable fair comparisons across different models. Therefore, we use **Mean Absolute Error (MAE)** and **Rooted Mean Squared Error (MSE)** as evaluation metrics. These metrics measure the error between the actual 12th point $p_{12}^i$ and the predicted 12th point, converted to a distance measure (i.e., meters).

Since MAE and MSE only provide a macro-level quantitative assessment, they may not capture the usability of predictions in certain applications. Often, only predictions with errors below a specific threshold are considered useful. To address this, we introduce a **pass rate metric**. We define $R_r$ as the proportion of predictions with errors less than $r$:

$$R_r = \frac{1}{N} \sum_{i=1}^{N} \mathbb{I}\left(\phi\left(p_{12}^i, \hat{p}_{12}^i\right) < r\right), \qquad (1)$$

where $\mathbb{I}(\cdot)$ is the indicator function, which equals 1 if the condition is satisfied and 0 otherwise, and the function $\phi$ converted the difference to a distance measure.

## 4. Experiment

### 4.1. Baselines

We selected four categories of models for our experiments.

**1. State-of-the-art VLMs:** These can be further divided into the following subcategories: **Inference-time scaling VLMs**, such as GeminiFlash2 Thinking 1219(Gemini Team, 2024b)(Guessed to be ∼175B), and QVQ-72B-Preview(Team, 2024). **General generative VLMs**, including Claude3.5-Sonet(∼175B)(Claude, 2024b), Claude3-Haiku(Guessed to be ∼8B)(Claude, 2024a), Qwen2VL-72B-Instruct(Wang et al., 2024), Qwen2VL-7B-Instruct(Wang et al., 2024), Pixtral-Large-2411(124B)(MistralAI, 2024), Pixtral-12B(Agrawal et al., 2024), GeminiPro-1.5(Guessed to be ∼175B)(Gemini Team, 2024a), GeminiFlash1.5-8B(Gemini Team, 2024a), GPT4o(∼200B)(OpenAI, 2024a), GPT4o-mini(∼8B)(OpenAI, 2024b), Llama3.2-90B-vision-Instruct(Llama, 2024), and Llama3.2-11B-

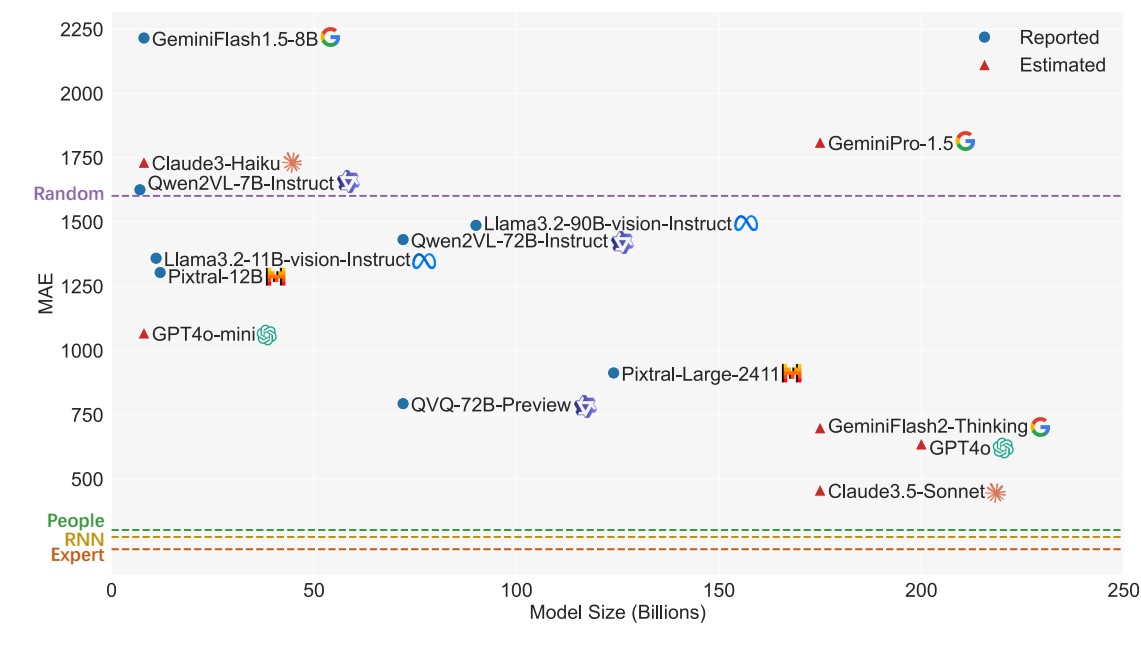

*Figure 4.* The performance of models benchmarked in VLMLocPredictor. The x-axis represents the model's parameter size, while the y-axis denotes the MAE across all datasets. Circular markers indicate models with publicly known parameter sizes, whereas triangular markers represent models whose parameter sizes are estimated either in this study or based on related literature.

vision-Instruct(Llama, 2024). The number of parameters of some models are derived from (Abacha et al., 2025). The *Instruct* in the model name may be omitted for simplicity. Because GPT-o1(OpenAI, 2024c) temporarily only supports their tier-5 users, we did not compare with it.

**2. Human performance**: We first collect individuals from around 100 volunteers, whose performance is referred to as the baseline **People**. Additionally, we engaged 20 domain experts specializing in trajectory analysis to predict the next trajectory point, with their performance serving as the baseline **Expert**.

**3. Random Baseline**: The baseline replicates the procedure of our VGLS but outputs blue and yellow randomly.

**4. Deep learning model**: As this is not the primary focus of our work, we selected the simplest model in this category, a single-layer **RNN**(Elman, 1990). **Moreover, this comparison is inherently unfair**, as RNN are extensively trained on urban trajectory data, whereas our evaluated models are general VLMs that have not undergone any specialized training for trajectory prediction.

### 4.2. Experimental Settings

Most VLMs were accessed via the OpenRouter API[4]. Especially, GeminiFlash2 Thinking was accessed through

Google AI Studio[5]. For the RNN model, we configured the hidden layer size to 64 with a single layer. The learning rate was selected from the range $[1e\text{-}4, 5e\text{-}4, 1e\text{-}3, 5e\text{-}3, 1e\text{-}2]$, with the optimal value being $1e\text{-}3$.

### 4.3. Overall Evaluation

#### 4.3.1. QUANTITATIVE COMPARISON

To provide a clear illustration, Figure 4 presents the overall MAE of all models across different settings. It is evident that the Gemini1.5 series, Qwen2 series, Claude 3-Haiku, and Llama 3.2 series exhibit either no or very weak capability in next-location prediction on maps. Notably, the Gemini 1.5 series performs significantly worse than even the Random baseline, despite the theoretical expectation that random blue and yellow guessing should be the worst-case scenario. This suggests that Gemini 1.5 may have an inherently confused understanding of map-based spatial relationships. In contrast, the Pixtral series, GPT-4o series, QVQ, and Claude 3.5 Sonet demonstrate a certain level of next-location prediction capability. Furthermore, as model size increases, we observe a clear scaling effect, indicating that larger models tend to perform better.

Additionally, Table 2 presents the complete experimental results. One striking observation is that ordinary humans

[4] https://openrouter.ai

[5] https://aistudio.google.com/

*Table 2.* The results benchmarked in VLMLocPredictor. The bolded values represent the best-performing results among all VLMs.

| Model Name | Easy | | | | | Medium | | | | | Hard | | | | |
|---|---|---|---|---|---|---|---|---|---|---|---|---|---|---|---|
| | MAE | RMSE | $R_{100}$ | $R_{500}$ | $R_{2000}$ | MAE | RMSE | $R_{100}$ | $R_{500}$ | $R_{2000}$ | MAE | RMSE | $R_{100}$ | $R_{500}$ | $R_{2000}$ |
| Expert | 136.09 | 154.56 | 38.89 | 100.00 | 100.00 | 205.25 | 244.81 | 22.73 | 97.73 | 100.00 | 309.40 | 345.57 | 5.56 | 91.67 | 100.00 |
| People | 236.45 | 245.15 | 0.00 | 100.00 | 100.00 | 340.93 | 366.93 | 0.00 | 90.91 | 100.00 | 420.62 | 443.49 | 0.00 | 80.56 | 100.00 |
| RNN | 119.23 | 134.91 | 41.67 | 100.00 | 100.00 | 235.80 | 272.13 | 18.18 | 95.45 | 100.00 | 338.94 | 396.53 | 11.11 | 75.00 | 100.00 |
| Random | 1471.20 | 1623.26 | 0.00 | 5.56 | 72.22 | 1869.11 | 2126.27 | 0.00 | 11.36 | 59.09 | 1799.79 | 2063.48 | 0.00 | 11.11 | 52.78 |
| Claude3.5-Sonnet | **395.11** | **456.33** | 8.33 | **66.67** | **100.00** | **411.31** | **492.60** | **6.82** | **65.91** | **100.00** | **574.31** | **634.87** | 5.56 | 33.33 | **100.00** |
| GPT4o | 593.12 | 672.79 | 5.56 | 41.67 | **100.00** | 615.57 | 697.32 | 4.55 | 43.18 | **100.00** | 705.17 | 820.04 | 2.78 | **36.11** | 97.22 |
| GeminiFlash2-Thinking | 616.94 | 851.46 | **16.67** | 55.56 | 94.44 | 674.08 | 834.34 | 4.55 | 38.64 | 95.45 | 808.84 | 1097.23 | **5.56** | 41.67 | 94.44 |
| Pixtral-Large-2411 | 639.85 | 773.43 | 8.33 | 47.22 | 97.22 | 912.35 | 1172.99 | 0.00 | 36.36 | 88.64 | 1180.27 | 1351.08 | 2.78 | 8.33 | 86.11 |
| QVQ-72B-Preview | 774.08 | 885.34 | 2.78 | 25.00 | 97.22 | 882.45 | 1164.98 | 2.27 | 27.27 | 95.45 | 1026.00 | 1418.25 | 0.00 | 22.22 | 86.11 |
| GPT4o-mini | 950.89 | 1113.79 | 0.00 | 16.67 | 97.22 | 1019.16 | 1333.31 | 2.27 | 29.55 | 88.64 | 1238.64 | 1467.57 | 0.00 | 8.33 | 80.56 |
| Llama3.2-11B-vision | 1155.59 | 1347.48 | 0.00 | 19.44 | 83.33 | 1579.42 | 1902.44 | 0.00 | 15.91 | 63.64 | 1301.34 | 1567.87 | 0.00 | 19.44 | 77.78 |
| Pixtral-12B | 1158.86 | 1434.75 | 0.00 | 25.00 | 80.56 | 1315.29 | 1720.98 | 0.00 | 15.91 | 81.82 | 1443.41 | 1720.40 | 0.00 | 19.44 | 69.44 |
| Qwen2VL-72B | 1212.56 | 1398.39 | 2.78 | 11.11 | 83.33 | 1485.76 | 1727.43 | 2.27 | 9.09 | 77.27 | 1591.81 | 1784.99 | 0.00 | 2.78 | 75.00 |
| Llama3.2-90B-vision | 1311.17 | 1451.96 | 0.00 | 11.11 | 83.33 | 1521.25 | 1737.67 | 0.00 | 4.55 | 75.00 | 1622.60 | 1810.81 | 0.00 | 8.33 | 72.22 |
| Qwen2VL-7B | 1381.97 | 1550.44 | 0.00 | 11.11 | 75.00 | 1514.28 | 1842.73 | 0.00 | 18.18 | 75.00 | 2000.92 | 2272.47 | 0.00 | 8.33 | 61.11 |
| GeminiPro-1.5 | 1485.57 | 1710.08 | 0.00 | 0.00 | 75.00 | 1889.97 | 2237.94 | 0.00 | 0.00 | 59.09 | 2029.69 | 2424.87 | 0.00 | 5.56 | 66.67 |
| Claude3-Haiku | 1674.99 | 1840.93 | 0.00 | 0.00 | 66.67 | 1753.63 | 2133.02 | 0.00 | 2.27 | 63.64 | 1779.91 | 2015.82 | 0.00 | 2.78 | 66.67 |
| GeminiFlash1.5-8B | 1792.72 | 1945.49 | 0.00 | 0.00 | 63.89 | 2178.22 | 2482.96 | 0.00 | 0.00 | 52.27 | 2671.60 | 2965.60 | 0.00 | 0.00 | 33.33 |

fail to achieve an accurate prediction error below 100 meters on any category of trajectory tasks. However, certain VLMs successfully reduce the prediction error to below 100 meters, suggesting that VLMs possess some level of intelligence in predicting unknown future locations. However, human consistently keep most trajectory errors below 500 meters, with all errors below 2000 meters, demonstrating a strong commonsense reasoning capability in spatial intelligence. Among the VLMs, only Claude 3.5 Sonet is capable of keeping all trajectory errors under 2000 meters. Moreover, in terms of accuracy within the 500-meter range, there remains a considerable gap between human performance and that of current VLMs, indicating that most models still lack essential commonsense intelligence in spatial understanding.

Furthermore, as the task difficulty increases, more models perform worse than even the Random baseline. For instance, GPT-4o-mini performs comparably to Random in Easy and Medium settings, but in the Hard setting, its $R_{500}$ score even falls below that of Random, suggesting its limitations in comprehending complex road networks. In such challenging scenarios, only the three largest models achieve satisfactory performance, implying that model size plays a critical role in understanding intricate maps.

Although VLMs have not yet surpassed RNNs, it is important to note that **not all scenarios have sufficient data for training RNNs.** In contrast, the VLMs examined in this study serve as a training-free approach to next-location prediction, demonstrating the ability to adapt to various scenarios **without the need for task-specific training**.

### 4.3.2. VLM V.S. HUMAN

By now, our results indicate that none of the current VLMs outperform human performance. However, it is evident

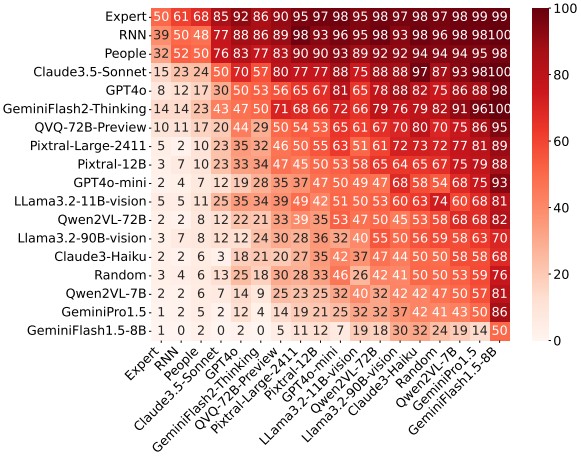

*Figure 5.* **The win rate between baselines.** The value in the i-th row and j-th column represents the percentage of scenarios in which the i-th model outperforms the j-th baseline.

that with increasing model scale, the performance of larger VLMs is approaching that of humans. **This suggests that, in the near future, VLMs may reach human-level performance averages.**

We also conducted experiments comparing the win rates between baselines as shown in Figure 5. The win rate for each pair of models was determined by evaluating the error comparison for each trajectory: for any two models, the model with the smaller error was considered the winner. The average win rate across the entire dataset was then computed. The error for humans was represented as the average human error.

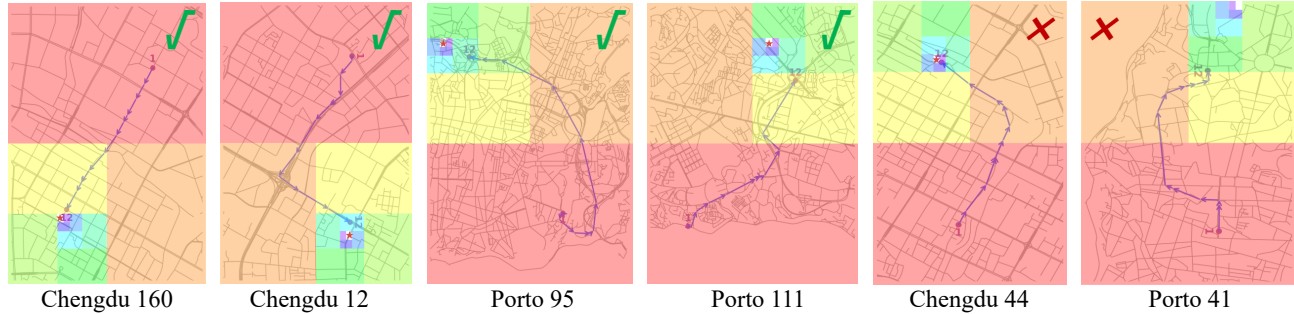

|        |         |         |          |         |         |
| Chengdu 160 | Chengdu 12 | Porto 95 | Porto 111 | Chengdu 44 | Porto 41 |

*Figure 6.* **Case Study**. The examples are all taken from the predictions of Claude3.5 Sonet. The first four represent reasonable predictions, while the last two correspond to less reasonable predictions. The colored regions indicate areas that the model has excluded from consideration, and the red star marks the model's final predicted location.

From this experiment, we observe that experts outperform all models in terms of accuracy. However, it is promising that in 24% of cases, the best-performing VLM outperforms the average human prediction. Moreover, in 23% of cases, a VLM that was **not specifically fine-tuning on trajectory data** outperforms an RNN that was **pre-trained on such data**, demonstrating their potential to surpass models specialized in trajectory prediction.

### 4.4. Ablation Study

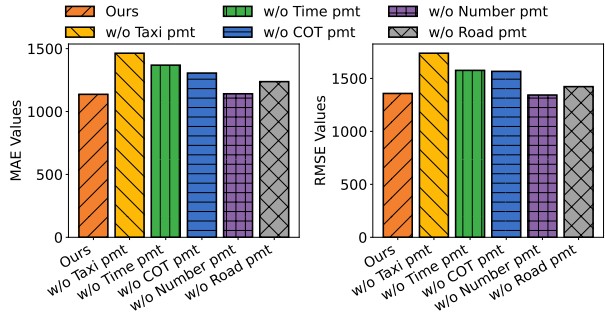

*Figure 7.* **The illustration of ablation study.** This experiment validates that all the proposed prompts exhibit a certain degree of effectiveness. The notation w/o denotes the removal of a specific prompt.

In Figure 7, we conduct a comprehensive ablation study on the Pixtral12b model using the Porto dataset to validate the effectiveness of each prompt in our method. In this experiment: **Ours** refers to using the full set of prompts. **w/o taxi pmt** removes the prompt describing the trajectory as that of a taxi driver. **w/o Time pmt** removes the prompt indicating that each trajectory point corresponds to a 45-second interval. **w/o COT pmt** removes the chain-of-thought reasoning prompt. **w/o Number pmt** removes the prompt specifying

that point 1 is the first and point 12 is the twelfth in the trajectory. **w/o Road pmt** removes the prompt describing the road network. The Pixtral12b model was chosen for its optimal balance of performance and computational cost.

Our results show that the full-prompt configuration (Ours) achieves the best MAE across all setups. Furthermore, we observe the following: The most impactful prompt is the taxi pmt. This likely indicates that explicitly constraining the trajectory to the road significantly reduces the possible trajectory space. The Time pmt, Road pmt, and COT pmt all contribute significantly to performance, suggesting that providing auxiliary information is effective in improving model predictions. Although the w/o Number pmt configuration shows slightly better results in RMSE, the MAE metric provides a more intuitive measure of prediction quality. Therefore, we retain the Number pmt in our final design.

### 4.5. Case Study

In this section, we present a visual analysis of selected prediction results from Claude3.5 Sonet. The first four examples illustrate reasonable predictions, while the last two exhibit less reasonable outcomes. More examples including textual outputs can be found in the Appendix D.1.

First, in the Chengdu160 trajectory, which features near-uniform motion, the model accurately predicts the next location along the uniform trajectory, demonstrating its inherent spatial reasoning capabilities. Similarly, in the Chengdu12 trajectory, the model successfully identifies a feasible turning point after a curve, suggesting that it has learned a reasonable understanding of the topological relationships within the road network. In more complex scenarios, such as Porto95 and Porto111, the model effectively selects a viable next location based on both the trajectory and road network constraints, further supporting its capability for next-location prediction.

However, the model's predictions can sometimes be influenced by previous trajectory patterns. For instance, in Chengdu44, the movement from the fourth to the fifth point is minimal, leading the model to predict a stop at the 13th point. Conversely, in Porto41, where the displacement between the 6th and 7th points is large, the model overestimates the movement range when predicting the 13th point. While these latter predictions may appear suboptimal, they still indicate that the model is capturing historical trajectory patterns. This suggests that its predictive capabilities can be further improved with additional refinement and training.

### 4.6. The Accuracy in Each Step

To evaluate the step-wise accuracy of our proposed method and assess whether the progressively smaller colored regions in later steps affect the results, we conducted a comprehensive experiment on six baseline models using the Chengdu Dataset. At each step $i$, the model was provided with images where the first $i-1$ steps were correct, allowing us to isolate and assess its accuracy at step $i$.

Our findings indicate that Claude 3.5 Sonnet aligns well with our expectations. The model demonstrates high accuracy when the prediction region is large, as determining the next location within a broad area is relatively straightforward (e.g., distinguishing between left and right hemispheres). However, as the prediction region shrinks, the accuracy gradually declines, ultimately approaching 50% in the final steps, where multiple directions may be equally plausible. Nevertheless, due to constraints imposed by road networks and spatial structure, its accuracy remains slightly above 50%, suggesting a strong capability to interpret maps and perceive fine-grained spatial information.

In contrast, models such as GPT-4o experience a sharp accuracy drop beyond the eighth step, with performance rapidly approaching or even falling below 50%, indicating difficulties in processing small-scale visual inputs. Moreover, Gemini Flash 1.5-8B and Qwen2VL-7B exhibit accuracy fluctuations of around 50% throughout all steps, suggesting that these models may lack the inherent capability to directly interpret map-based spatial data, which ultimately results in subpar performance.

### 5. Related Work

Due to the space limit, we only provide related works on next location prediction. The full version of related work can be found in the Appendix C.

In this paper, we define the Next Location Prediction task as predicting the next location point given a sequence of past trajectory points sampled at regular intervals. Historically, early approaches modeled dynamic relationships in human mobility using Markov chains(Norris, 1998), but these mod-

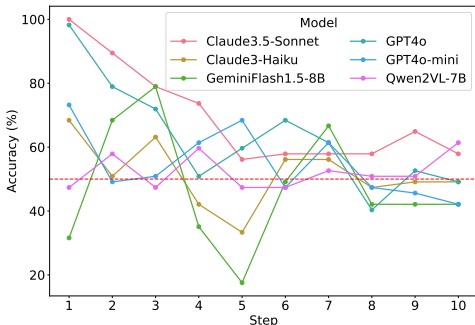

*Figure 8.* **The illustration of the accuracy in each step.** Only Claude 3.5 Sonet demonstrates relatively stable and strong predictive performance. In contrast, all other models exhibit instability or produce results that tend to be random when dealing with smaller-scale images.

els were limited to first-order dependencies like (Gao et al., 2019; Wang et al., 2021). With the advent of deep learning, recurrent neural networks (RNNs)(Elman, 1990) became popular for next-point prediction, yielding promising results such as (Chen et al., 2023; Feng et al., 2022a;b). Recently, large language models (LLMs) have been investigated for leveraging semantic information along the journey to further improve location predictions. The representative works are LLM-Mob(Wang et al., 2023b) and Agent-Move(Du et al., 2024). However, these approaches still face challenges related to cross-city transferability. **While some language-based models have demonstrated a degree of cross-city adaptation, this often requires fine-tuning LLMs, which does not fully showcase their generalization capabilities.**

### 6. Conclusion

In this work, we explored the capability of VLMs to predict the next location point on map data through our benchmark, VLMLocPredictor. To this end, we proposed the Visual Guided Location Search (VGLS) framework, which equips VLMs with the ability to predict subsequent trajectory points. We evaluated this framework by constructing datasets and conducting human experiments. Our findings demonstrate that VLMs possess a certain degree of predictive capability for next-location tasks. Although current VLMs still fall short of the performance achieved by RNNs and human predictions, our experiments reveal that VLMs surpass human predictions in 24% of scenarios and outperform experts in 15% of scenarios. Given that Large Language Models (LLMs) struggle to process complex map data, we believe that our findings pave the way for the development of more effective next-location prediction models.

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

# A. Prompt

Hello, you are an expert in next-location prediction.

### Known Information:

1. This trajectory represents the movement of a taxi driver.

2. Each arrow represents the distance and direction of movement over approximately 45 seconds.

3. The 11 arrows in the diagram show the sequential movement from the first trajectory point to the 12th trajectory point.

4. The starting position is marked with a purple dot labeled as point 1 in the diagram, and the current position is marked with a purple dot labeled as point 12.

5. Gray lines in the diagram represent drivable roads, while white areas indicate buildings or other obstacles.

### Question:

Based on the following requirements, determine whether the next position (the 13th trajectory point) approximately 45 seconds later from the current purple dot position (the 12th trajectory point) is more likely to be in the blue region or the yellow region.

### Requirements:

1. Return the answer in JSON format, containing a key `ANS`. If the final answer is the blue region, the value should be `0`; if the final answer is the yellow region, the value should be `1`.

2. Let's think step by step.

*Figure 9.* Our prompt.

# B. Further Experiment

## B.1. Do different situations affect model performance?

In this experiment, we investigate whether different contextual factors influence model performance, which in turn determines the classification of tasks into easy, medium, and hard. We consider six potential factors: 1. The cumulative distance from the first point to the twelfth point ($D_{1,12}$). 2. The distance between the twelfth and thirteenth points ($D_{12,13}$). 3. The cumulative rotation angle from the first point to the twelfth point ($A_{1,12}$). 4. The rotation angle between the twelfth and thirteenth points ($A_{12,13}$). 5. The number of roads present in the map (Road Count). 6. The geographical area covered (Area Square).

For each of these factors, we compute its Pearson correlation coefficient with human prediction errors, as well as the corresponding p-value. The Pearson correlation coefficient quantifies the linear relationship between two variables, while the p-value measures the statistical significance of the correlation.

*Table 3.* PCC and Significance for Different Metrics

| Metric | PCC | p-value |
|---|---|---|
| $D_{1,12}$ | 0.16 | 0.09 |
| $D_{12,13}$ | 0.21 | 0.03 |
| $A_{1,12}$ | 0.16 | 0.08 |
| $A_{12,13}$ | 0.00 | 0.98 |
| Road Count | 0.21 | 0.02 |
| Area Square | 0.18 | 0.06 |

Our findings reveal that both $D_{12,13}$ and Road Count exhibit a positive correlation (PCC$> 0.2$ and p-value $< 0.05$) with prediction errors, suggesting that these two factors play a crucial role in determining task difficulty. Based on these results,

we classify the dataset into easy, medium, and hard categories according to the methodology described earlier, using $D_{12,13}$ and Road Count as the primary indicators.

### B.2. Choice of Color Combinations

We also investigated the impact of two specific color pairs on the model's performance in image-based trajectory prediction. Six different color pairs were tested: Blue-Green, Blue-Yellow, Green-Yellow, Red-Blue, Red-Yellow, and Red-Green. These tests were conducted using the Pixtral12b model on the Porto dataset.

Our findings reveal that the Blue-Yellow color pair yields the best results across all models. Consequently, in the previous experiments, we chose Blue and Yellow as the distinguishing colors. This result is somewhat surprising, as one might expect color pairs with stronger contrast, such as Red-Blue and Red-Green, to perform better. However, these high-contrast color pairs resulted in poorer performance, suggesting that contrast might not always enhance model accuracy and could even negatively affect the model's decision-making process.

*Table 4.* Performance of Different Color Combinations for Annotation

| Color Combination | MAE | RMSE |
| --- | --- | --- |
| Blue-Green | 1268.26 | 1483.99 |
| Blue-Yellow | **1136.82** | **1357.96** |
| Green-Yellow | 1366.23 | 1626.04 |
| Red-Blue | 1578.96 | 1972.51 |
| Red-Yellow | 1469.21 | 1737.07 |
| Red-Green | 1613.08 | 1978.04 |

## C. Full Related Work

### C.1. Vision-Language Model

Vision-language models (VLMs) are capable of reasoning through the input of both text and images. Inspired by the success of large language models such as GPT(Brown et al., 2020), Claude(Anthropic, 2023), and LLaMA(Dubey et al., 2024), which many believe mark the advent of AGI(Bubeck et al., 2023), researchers have naturally begun exploring whether general-purpose models can also understand images. These models leverage large-scale datasets containing paired image-text samples, enabling them to capture rich semantic relationships between modalities.

VLMs can broadly be categorized into two main types based on their underlying methodology. The first category includes models such as CLIP(Radford et al., 2021), GLIP(Li et al., 2022), which rely on contrastive learning to generate representations by aligning image and text embeddings in a shared space. However, these models are not capable of generating texts. The second category comprises generative models such as GPT4 series(OpenAI, 2024a;b), Gemini series(Gemini Team, 2024b;a), Pixtral (Agrawal et al., 2024; MistralAI, 2024), and Claude series(Claude, 2024a;b; Anthropic, 2023). These models go beyond simple alignment, aiming to generate coherent and contextually rich outputs conditioned on multi-modal inputs. By employing transformer-based architectures(Vaswani et al., 2017), these generative VLMs excel in complex reasoning and content generation tasks, making them well-suited for applications requiring in-depth understanding and generation of multi-modal data.

In this work, we focus on generative vision-language models, leveraging their capability to model complex spatial and temporal relationships. However, while VLMs have already shown excellent performance across most datasets, fundamental visual intelligence required for survival in many animals, such as Next Location Prediction, remains untested by existing benchmarks.

### C.2. Next Location Prediction

In this paper, we define the Next Location Prediction task as the problem of predicting the next location point given a sequence of past trajectory points sampled at regular intervals. This task represents a fundamental aspect of biological intelligence. For example, cats predict the next point in a mouse's trajectory to catch it, and humans predict the next position of nearby vehicles to avoid collisions. The focus of this work is on predicting the next location for taxis, a task of significant

real-world importance. Accurate predictions can lead to reduced waiting times, improved route optimization, and better resource allocation within transportation networks.

Historically, early approaches modeled dynamic relationships in human mobility using Markov chains(Norris, 1998), but these models were limited to first-order dependencies like (Gao et al., 2019; Wang et al., 2021). With the advent of deep learning, recurrent neural networks (RNNs)(Elman, 1990) became popular for next-point prediction, yielding promising results such as (Chen et al., 2023; Feng et al., 2022a;b). Subsequently, Graph-based models and geographic feature data were explored to enhance next-point prediction in works like (Li et al., 2021; Yang et al., 2022). Recently, large language models (LLMs) have been investigated for leveraging semantic information along the journey to further improve location predictions. The representative works are LLM-Mob(Wang et al., 2023b) and Agent-Move(Du et al., 2024). However, these approaches still face challenges related to cross-city transferability. While some language-based models have demonstrated a degree of cross-city adaptation, this often requires specialized LLMs, which do not fully showcase their generalization capabilities.

# D. Selected Dialogue Records

In this section, we present a complete ten step dialogue with the VLM, illustrating the iterative reasoning process. Additionally, we showcase several failure cases where the VLM makes incorrect predictions. In these cases, the erroneous regions are highlighted in red to clearly indicate the model's mistakes.

## D.1. A Complete Dialogue Procedure

### User

Hello, you are an expert in next-location prediction …
determine whether the next position … is more likely to
be in the blue region or the yellow region … [See the
full version in Appendix A]

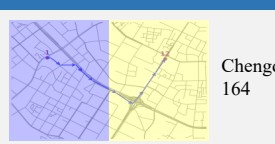

Chengdu
164

### VLM (Round 1)

Let me analyze this step by step:
1. Trajectory Pattern Analysis:
- The taxi shows a consistent directional movement from point 1 to point 12
- The movement appears to be following a major road
- The arrows indicate relatively steady speed and direction
2. Current Position Analysis:
- Point 12 is located in the yellow region
- The taxi appears to be on what looks like a main road
- The movement has been predominantly in one direction
3. Movement Characteristics:
- The taxi has maintained a fairly straight path
- There are no sharp turns in the recent trajectory
- The speed (arrow length) has been consistent
4. Road Network Analysis:
- The taxi is following what appears to be a major road
- The road continues in the same direction in the yellow region
- There are no immediate major intersections visible that would suggest a turn
5. Final Reasoning:
- Given the consistent straight movement
- The continuation of the road in the same direction
- The absence of any obvious reason to turn back
- The momentum and direction of travel
- The taxi is most likely to continue in the same direction within the yellow region
Therefore, the next position (13th point) is more likely to be in the yellow region.
{"ANS": 1}

## User

Hello, you are an expert in next-location prediction … determine whether the next position … is more likely to be in the blue region or the yellow region … [See the full version in Appendix A]

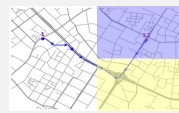

Chengdu
164

## VLM (Round 2)

Let me analyze this step by step:
1. Movement Pattern Analysis:
- The taxi starts from point 1 and follows a relatively consistent direction
- The arrows show a general diagonal movement from left to right
- The most recent arrows (leading to point 12) show upward movement into the blue region
2. Road Network Analysis:
- The current position (point 12) is on a road in the blue region
- There appear to be continuous road connections in both the blue and yellow regions
- The road network shows a grid-like pattern
3. Recent Trajectory Consideration:
- The last few movements show a clear trend moving upward
- The taxi has just completed a significant northward movement
- The momentum and direction suggest continued movement in this direction
4. Final Analysis:
- Given the recent upward trajectory
- Considering the road network allows for continued movement in this direction
- The momentum and pattern suggests the taxi is more likely to continue into the blue region
- The last few movements strongly indicate a destination in the blue region rather than reversing course to the yellow region
{"ANS": 0}

## User

Hello, you are an expert in next-location prediction … determine whether the next position … is more likely to be in the blue region or the yellow region … [See the full version in Appendix A]

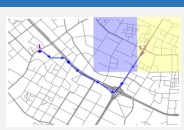

Chengdu
164

## VLM (Round 3)

Let me think about this step by step:
1. First, let's analyze the movement pattern:
- The taxi started from point 1 and moved consistently eastward
- After crossing a major intersection, it made a sharp turn northeast
- The movement has been relatively steady in terms of speed (arrow lengths)
- The taxi is currently at point 12, heading in a northeastern direction
2. Looking at the road network:
- The current position (point 12) is at the boundary of blue and yellow regions
- There are available roads continuing in both regions
- The taxi appears to be on a main road that continues into the yellow region
3. Considering momentum and direction:
- The last few arrows show a clear northeastern trajectory
- The natural continuation of this path would lead into the yellow region
- To enter the blue region would require a sharp turn to the left, which breaks the current movement pattern
4. Based on typical driving behavior:
- Drivers tend to continue in their current direction unless there's a specific reason to turn
- The road structure supports continuing straight into the yellow region
- The recent trajectory suggests a destination in the yellow region
5. Final analysis:
- Given the consistent northeastern movement
- The road network layout
- The momentum of the trajectory
- The most likely next position would be in the yellow region
{"ANS": 1}

## User

Hello, you are an expert in next-location prediction …
determine whether the next position … is more likely to
be in the blue region or the yellow region … [See the
full version in Appendix A]

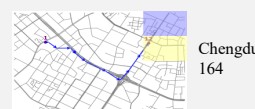

Chengdu
164

## VLM (Round 4)

Let me analyze this step by step:
1. Movement Pattern Analysis:
- The taxi has been moving in a generally eastward direction
- The movement shows a clear pattern of following the road network
- The arrows indicate relatively consistent speed and direction changes aligned with road structure
2. Current Position Analysis:
- Point 12 is located at the edge of both yellow and blue regions
- The taxi has just completed a turn and reached this intersection
- There are road options available in both regions
3. Road Network Analysis:
- Looking at the gray lines (roads), there are viable paths in both the blue and yellow regions
- The road network appears to be denser in the yellow region
- The yellow region seems to be a more developed area with more potential destinations
4. Momentum and Direction:
- The last few arrows show the taxi moving towards and into the yellow region
- The final arrow's direction suggests continued movement into the yellow region
- There's no indication of an immediate turn that would take it into the blue region
5. Final Reasoning:
- Given the taxi's current momentum and direction
- Considering the denser road network in the yellow region
- Taking into account the natural flow of the trajectory
- The taxi is more likely to continue into the yellow region rather than make a turn into the blue region
{"ANS": 1}

## User

Hello, you are an expert in next-location prediction …
determine whether the next position … is more likely to
be in the blue region or the yellow region … [See the
full version in Appendix A]

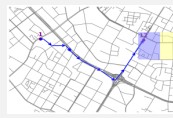

Chengdu
164

## VLM (Round 5)

Let me think about this step by step:
1. Trajectory Analysis:
- The taxi started from point 1 and moved consistently eastward
- After crossing a major intersection, it made a sharp turn towards the northeast
- The movement has been relatively steady and purposeful
2. Current Position Analysis:
- Point 12 is at the edge of the blue region
- The taxi appears to be moving into the blue region based on the last arrow
- The road network continues in both blue and yellow regions
3. Movement Patterns:
- The taxi has maintained consistent direction changes
- No sudden U-turns or erratic movements are observed
- The most recent movement shows entry into the blue region
4. Road Network Consideration:
- Both blue and yellow regions have accessible roads
- The taxi's current trajectory aligns more with the blue region's road network
- Continuing the current direction would naturally lead into the blue region
5. Final Analysis:
- Given the steady movement pattern
- The current direction heading into the blue region
- The natural flow of the road network
- The momentum of the trajectory
Based on these observations, the taxi is more likely to continue into the blue region rather than make a turn into the
yellow region.
{"ANS": 0}

## User

Hello, you are an expert in next-location prediction … determine whether the next position … is more likely to be in the blue region or the yellow region … [See the full version in Appendix A]

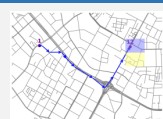

Chengdu
164

## VLM (Round 6)

Let me analyze this step by step:
1. Movement Pattern Analysis:
- The taxi has been moving in a generally consistent direction
- The arrows show a relatively smooth trajectory with few sharp turns
- The most recent movements are heading towards the northeast
2. Current Position (Point 12):
- Located at an intersection
- Has two main possible directions: continue straight (into blue region) or turn right (into yellow region)
3. Road Network Analysis:
- The road continues in both directions (blue and yellow regions)
- Both regions appear to be accessible from the current position
4. Movement Momentum:
- The last few arrows show a clear directional trend
- The taxi appears to be maintaining its forward momentum
- There's no indication of an upcoming turn
5. Final Analysis:
- Given the consistent forward movement pattern
- The momentum of the trajectory
- The lack of turning indicators in recent movements
- The natural continuation would be into the blue region
Based on these observations, the taxi is more likely to continue its forward momentum into the blue region rather than make a turn into the yellow region.
{"ANS": 0}

## User

Hello, you are an expert in next-location prediction … determine whether the next position … is more likely to be in the blue region or the yellow region … [See the full version in Appendix A]

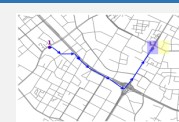

Chengdu
164

## VLM (Round 7)

Let me analyze this step by step:
1. Trajectory Pattern Analysis:
- The taxi has been moving in a generally consistent direction
- The movement shows a clear path from west to east
- The arrows indicate relatively uniform speed and direction
2. Current Position Analysis:
- Point 12 is at an intersection
- The taxi has just completed a turn towards the northeast
- There are two possible paths forward: straight ahead (yellow region) or continuing the turn (blue region)
3. Road Network Analysis:
- The road continues in both directions (into both blue and yellow regions)
- The yellow region represents continuing straight on the current trajectory
- The blue region would require maintaining the turning motion
4. Movement Physics:
- Given the taxi has just completed a turn
- It would be more natural to straighten out rather than continue turning
- The previous momentum suggests a straightening of trajectory
5. Decision Factors:
- Natural driving behavior typically straightens out after completing a turn
- The yellow region aligns better with the general east-bound trajectory
- The road network allows for straightforward movement into the yellow region
Based on these observations, the taxi is more likely to continue straight into the yellow region rather than continue turning into the blue region.
{"ANS": 1}

935
936
937
938
939
940
941

## User

Hello, you are an expert in next-location prediction … determine whether the next position … is more likely to be in the blue region or the yellow region … [See the full version in Appendix A]

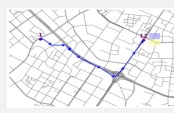

Chengdu 164

## VLM (Round 8)

Let me analyze this step by step:
1. Movement Pattern Analysis:
- The taxi has been moving in a generally consistent direction
- The arrows show a relatively straight path with few major turns
- The most recent movements show the taxi heading towards the intersection where the blue and yellow regions are located
2. Road Network Analysis:
- The current position (point 12) is at an intersection
- There are two possible paths: one leading to the blue region and one leading to the yellow region
- Both regions appear to be accessible via the road network
3. Movement Momentum:
- The recent trajectory shows the taxi has been maintaining a fairly straight path
- The last few arrows indicate a slight tendency towards the direction of the yellow region
- Given the momentum and direction of travel, continuing straight ahead seems more likely
4. Distance and Time Consideration:
- Given that each movement takes about 45 seconds
- The distance to both regions appears similar
- The yellow region would require less turning and maintain the current direction
5. Final Analysis:
- Based on the momentum of movement
- The straightforward path ahead
- The natural flow of the trajectory
- The yellow region appears to be the more likely next position
{"ANS": 1}

## User

Hello, you are an expert in next-location prediction … determine whether the next position … is more likely to be in the blue region or the yellow region … [See the full version in Appendix A]

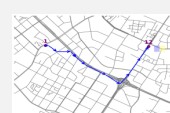

Chengdu 164

## VLM (Round 9)

Let me analyze this step by step:
1. Movement Pattern Analysis:
- The taxi has been moving in a generally consistent direction
- The trajectory shows a smooth path following the road network
- The arrows indicate steady movement with similar distances
2. Road Network Analysis:
- The taxi is currently at point 12
- There are two possible paths ahead: one leading to the blue region and one leading to the yellow region
- Both regions are accessible via the road network
3. Momentum and Direction Analysis:
- The last few arrows show the taxi moving upward and slightly to the right
- The taxi's current momentum and direction align more naturally with continuing straight ahead
- The turn required to reach the yellow region would be less natural given the current trajectory
4. Distance Consideration:
- Both regions appear to be within reasonable distance for a 45-second movement
- The blue region would require less deviation from the current path
- The yellow region would require a sharp turn to the right
5. Final Assessment:
- Given the taxi's consistent movement pattern
- Considering the natural flow of traffic
- Taking into account the momentum and current direction
- The blue region appears to be the more likely next location
{"ANS": 0}

990
991
992
993
994
995
996
997
998
999
1000
1001
1002
1003
1004
1005
1006
1007
1008
1009
1010
1011
1012
1013
1014
1015
1016
1017
1018
1019
1020
1021
1022
1023
1024
1025
1026
1027
1028
1029
1030
1031
1032
1033
1034
1035
1036
1037
1038
1039
1040
1041
1042
1043
1044

## User

Hello, you are an expert in next-location prediction …
determine whether the next position … is more likely to
be in the blue region or the yellow region … [See the
full version in Appendix A]

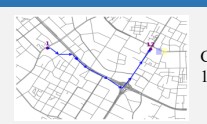

Chengdu
164

## VLM (Round 10)

Let me analyze this step by step:
1. Looking at the trajectory pattern:
   - The taxi starts from point 1 and moves in a generally eastward direction
   - The movement shows a clear purpose, following the road network
   - The arrows indicate relatively consistent speed and direction changes that align with the road structure
2. Analyzing the current position (point 12):
   - The taxi is at an intersection
   - There are two immediate possible directions: straight ahead (blue region) or right turn (yellow region)
   - The taxi has just completed a relatively straight movement to reach this point
3. Considering road network:
   - The blue region continues along the current road
   - The yellow region requires a right turn
   - Both options are valid paths according to the road network
4. Pattern analysis:
   - The previous movements show a preference for maintaining direction when possible
   - The taxi has already made its major turns to reach the destination area
   - The momentum and direction suggest continuing straight ahead
5. Conclusion:
   - Given the consistent movement pattern
   - The preference for maintaining direction
   - The natural flow of traffic
   - The blue region appears more likely as the next position
{"ANS": 0}

## D.2. The Dialogue of Wrong Cases

### User

Hello, you are an expert in next-location prediction …
determine whether the next position … is more likely to
be in the blue region or the yellow region … [See the
full version in Appendix A]

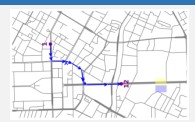

Chengdu
22

### VLM (Round 8, GPT-4o)

To predict the next position of the taxi driver (the 13th trajectory point), let's analyze the current trajectory and road options:

1. **Current Movement Pattern:**
   - The trajectory shows a consistent movement along the main road, moving southward from point 1 to point 12.
2. **Road Analysis from Point 12:**
   - The road from point 12 continues straight south, potentially leading towards the blue and yellow regions.
   - There is no immediate intersection or turn visible that would suggest a deviation from the current southward direction.
3. **Proximity to Regions:**
   - **The blue region is directly south of the current position and aligns more closely with the road trajectory.**
   - **The yellow region seems to be slightly off the main road and more to the southwest.**
4. **Direction and Likelihood:**
   - **Given the consistent pattern of southward movement, it is likely that the taxi will continue southward into the blue region.**

Based on this reasoning, the next position is more likely to be in the blue region.

{
  "ANS": 0
}

## User

Hello, you are an expert in next-location prediction … determine whether the next position … is more likely to be in the blue region or the yellow region … [See the full version in Appendix A]

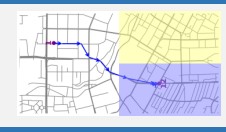

Chengdu
83

## VLM (Round 2, GPT4o)

To predict the next location, let's analyze the trajectory and road layout:

1. **Current Movement Direction**: The trajectory from point 1 to point 12 is predominantly southward.

2. **Available Roads**: The taxi is currently on a road heading directly south. **The road continues into the yellow region.**

3. **Road Layout**: The road layout suggests a continuation into the yellow region, as the blue region would require a left turn, which seems less aligned with the current trajectory.

4. **Obstacles**: **There are no visible road connections leading directly into the blue region** without deviating from the current path.

Given these observations, the taxi is more likely to continue south into the yellow region.
{ "ANS": 1}

## User

Hello, you are an expert in next-location prediction … determine whether the next position … is more likely to be in the blue region or the yellow region … [See the full version in Appendix A]

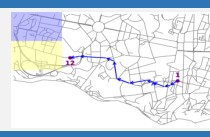

Porto
10

## VLM (Round 4, GPT4o)

Response:
 To determine whether the next position is more likely to be in the blue region or the yellow region, let's analyze the trajectory step by step:

1. **Trajectory Analysis**: The arrows indicate a clear pattern of movement primarily in the north-west direction from point 1 to point 12.

2. **Current Position**: At point 12, the trajectory is moving northwest along the road.

3. **Road Layout**: The road network suggests that continuing in the northwest direction aligns with the roads leading into the blue region.

4. **Region Proximity**: **The blue region is in the direct path of the continuation of the current trajectory, while the yellow region is further north and might require a change in direction.**

5. **Prediction**: Given the consistent directional movement and road alignment, it is more likely that the next trajectory point will fall within the blue region.

Therefore, the prediction for the next position, approximately 45 seconds later, is as follows:

{"ANS": 0}

## User

Hello, you are an expert in next-location prediction …
determine whether the next position … is more likely to
be in the blue region or the yellow region … [See the
full version in Appendix A]

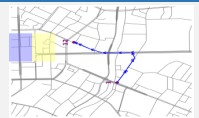

Chengdu
95

## VLM (Round 5, Claude3.5 Sonet)

Let me analyze this step by step:
1. Looking at the trajectory pattern:
  - The taxi started from point 1 and moved eastward
  - Then took a sharp turn northward
  - Has been moving consistently northward for several segments
  - Currently at point 12, still moving in a northward direction
2. Analyzing the momentum:
  - The last few arrows show a strong and consistent northward movement
  - The taxi appears to be traveling along a main road
  - There's no indication of slowing down or turning
3. Looking at the road network:
  - The taxi is on what appears to be a major north-south road
  - This road continues northward into both regions
  - **The blue region is directly north of the current position**
  - **The yellow region is slightly northeast/northwest of the current position**
4. Considering probable behavior:
  - Given the consistent northward movement
  - The direct continuation of the road into the blue region
  - The momentum of the trajectory
  - **The taxi is more likely to continue straight north**
5. Conclusion:
  - Based on the trajectory's momentum and road network
  - The blue region (directly north) is the more likely next location
  - It would be the natural continuation of the current movement pattern
{"ANS": 0}

## User

Hello, you are an expert in next-location prediction …
determine whether the next position … is more likely to
be in the blue region or the yellow region … [See the
full version in Appendix A]

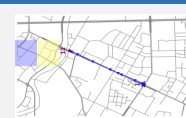

Chengdu
71

## VLM (Round 5, Claude3.5 Sonet)

Let me analyze this step by step:
1. Movement Pattern Analysis:
- The taxi has been moving consistently northward for the entire trajectory
- The arrows show a very straight path with minimal deviation
- The movement speed (arrow length) appears relatively constant
2. Current Position Analysis:
- Point 12 is at an intersection
- The taxi has just arrived at this junction
- There are multiple possible directions to turn from this point
3. Road Network Analysis:
- The yellow region has more complex road networks and intersections
- The blue region appears to have simpler, more direct routes
- Both regions are immediately accessible from the current position
4. Momentum Consideration:
- Given the consistent northward movement, there's a higher probability of continuing in the same general direction
- The momentum and direction suggest movement toward the blue region is more likely
5. Decision Factors:
- Historical trajectory shows strong directional consistency
- **The blue region aligns with the established movement pattern**
- The immediate road layout allows for continuation in this direction
- The 45-second timeframe would likely keep the taxi within the marked regions
Based on these observations, particularly the strong northward momentum and consistent movement pattern, the next
position is more likely to be in the blue region.
{"ANS": 0}

