# OpenReview forum: "Evaluating VLMs' General Ability on Next Location Prediction"
_ICML.cc/2025/Conference — Submitted to ICML 2025_

### Official Review · Reviewer_JUid · 2025-03-11

**Overall Recommendation:** 3

**Summary:**

This paper introduces a benchmark for evaluating the performance of vision-language models (VLMs) on next-location prediction. The benchmark is created with open-source map public taxi trajectory data. They draw the first 12 points of the taxi trajectory on the map and ask the VLMs to predict the location of the 13th point.

Using this benchmark, the authors evaluated the performance of 14 VLMs. They found the VLMs can produce meaningful predictions rather than random guesses.

The authors have also set up a platform to evaluate human prediction performance on this benchmark. They found there is a significant gap between the performance of VLMs and human prediction performance.


## update after rebuttal

No updates.

**Claims And Evidence:**

* The benchmark and study results are useful contributions to the community.

* However, it is up for debate how much value this new benchmark provides and what types of new research can benefit from it.

**Essential References Not Discussed:**

N/A

**Experimental Designs Or Analyses:**

* The experimental designs are sound.

**Methods And Evaluation Criteria:**

* The evaluation setup is reasonable.

**Other Comments Or Suggestions:**

N/A

**Other Strengths And Weaknesses:**

N/A

**Questions For Authors:**

N/A

**Relation To Broader Scientific Literature:**

N/A

**Theoretical Claims:**

* There are no theoretical proofs in the paper.

---

> ### Author Rebuttal · Authors · 2025-03-31
>
> Thank you for your valuable comments. We appreciate the time and effort you have devoted to reviewing our work. Below, we address your concerns in detail.
>
> **How much value does this new benchmark provide?**
>
> 1. **From the perspective of large vision-language models (VLMs)**:
>    The proposed benchmark is fundamentally designed to evaluate the *reasoning capabilities* of VLMs. Consider a simple example: suppose the vertices of a square are labeled A, B, C, and D in order. If a driver travels from A to B to C, and all roads are accessible without complex traffic rules, it is highly unlikely that the next point will be D—since a direct route from A to D would have been more efficient. This suggests that the prediction of the next location relies on understanding the *topological structure of the road network*. In more complex real-world maps, this task requires even stronger reasoning skills. Our benchmark provides a framework to quantitatively assess whether VLMs possess such abilities. As shown in our results, current VLMs still fall short of human-level map reasoning and spatial understanding.
> 2. From the perspective of next-location prediction:
>    Most existing approaches rely on learning city-specific trajectory patterns, often limiting their generalizability. In contrast, our work explores the potential of VLMs to leverage visual understanding of road networks to achieve generalizable next-location prediction. Because the visual representation of road networks tends to be domain-agnostic, VLM-based approaches offer the potential to build universal models for trajectory prediction. Our benchmark thus serves as a standardized evaluation framework for such vision-based next-location prediction, not limited to VLMs alone.
>
> **What types of new research can benefit from this benchmark?**
>
> 1. **Spatial computing:**
>    One promising application area is spatial computing. Real-world navigation often requires path planning over maps. Using VLMs to understand spatial layouts is an emerging research direction. Our benchmark allows researchers to quantitatively assess a model’s ability to reason about spatial structures and path planning, helping to avoid potential pitfalls in real-world deployments.
> 2. VLM-based reasoning:
>    As discussed above, the benchmark evaluates the reasoning abilities of VLMs. Since road networks form complex topological graphs, predicting the next step in a trajectory requires non-trivial reasoning over these structures. Our benchmark provides a testbed to evaluate whether VLMs can exhibit human-like map-based reasoning.
> 3. **General-purpose next-location prediction**:
>    By leveraging the general visual understanding capabilities of VLMs, there is strong potential to build general-purpose next-location predictors that are not bound to a specific city or dataset. As one of the first works to explore vision-based next-location prediction, we hope this study inspires further research into broadly applicable, vision-driven trajectory modeling.
>
> Once again, thank you for your thoughtful feedback and the time you’ve dedicated to our work. If you have any further questions or if any concerns remain unaddressed, please feel free to reach out, we would be happy to continue the discussion.
>
> Sincerely,
> The authors of Paper 740

---

> > ### Comment · Reviewer_JUid · 2025-04-04
> >
> > Thank you for your response.

---

### Official Review · Reviewer_Lpbt · 2025-03-15

**Overall Recommendation:** 2

**Summary:**

This paper explores the general capability of Vision-Language Models (VLMs) in performing next-location prediction, a key aspect of spatial intelligence that humans often handle through visual estimation. The authors introduce VLMLocPredictor, a novel benchmark designed to evaluate VLMs' predictive capabilities on next-location tasks. The paper makes the following key contributions:
1.	Visual Guided Location Search (VGLS) Module – A recursive refinement strategy that leverages visual guidance to iteratively narrow down the search space for next-location prediction. The VGLS module employs a hierarchical question-answering process where the VLM predicts which half of the map is more likely to contain the next location, progressively refining the prediction area through iterative feedback.
2.	Comprehensive Vision-Based Dataset – The dataset integrates open-source map data with publicly available taxi trajectory data from the Porto and Chengdu datasets. The dataset is categorized into easy, medium, and hard subsets based on the number of roads and trajectory distances, creating a structured evaluation framework.
3.	Human Benchmark – The authors established a human performance benchmark through a large-scale social experiment, where over 100 participants predicted the next trajectory point on the same test set used for the VLMs, generating over 10,000 samples.


## update after rebuttal
Thanks for providing rebuttals. Unfortunately, most of my concerns remain unaddressed. I would like to maintain my original rating.

**Claims And Evidence:**

In Section 4.3.1, the authors claim that "not all scenarios have sufficient data for training RNNs." However, the reviewer did not find concrete examples or real-world cases in the paper where training data for next-location prediction tasks is insufficient. This weakens the strength of the claim due to the lack of supporting evidence.
In Section 5, the authors state that methods based on large language models “face challenges related to cross-city transferability." However, no clear supporting evidence, such as cross-city experiments or quantitative analysis, is provided to substantiate this claim. Including such evidence would strengthen the credibility of the conclusion.

**Essential References Not Discussed:**

The paper provides a relatively comprehensive summary of related works. The authors have covered key references in the fields of next-location prediction, vision-language models (VLMs), and spatial reasoning. The discussion effectively situates the contributions within the broader scientific context, with no significant gaps in referencing essential prior works.

**Experimental Designs Or Analyses:**

The experimental design and analysis in the paper are generally well-constructed and thoughtfully aligned with the research objectives. However, a few aspects require further clarification or improvement:
1.	The paper reports that human performance was collected through a large-scale social experiment. However, it is unclear whether participants were provided with consistent task instructions or whether there were controls in place to ensure that human performance data is reliable and consistent across different scenarios. Furthermore, the paper compares VLM performance with that of "experts" without clarifying the qualifications or selection criteria for these experts. Providing more details about the expertise level of these participants would strengthen the validity of the comparison.
2.	The use of MAE, RMSE, and pass rate (for 100m, 500m, and 2000m thresholds) is appropriate for evaluating prediction accuracy and usability. However, the paper does not clarify the rationale behind selecting these specific distance thresholds for pass rate evaluation. An explanation based on the practical implications of these thresholds would make the evaluation more convincing.

**Methods And Evaluation Criteria:**

Overall, the proposed methods and evaluation criteria are thoughtfully designed and appropriately matched to the nature of the next-location prediction task. However, additional clarification on the stopping criteria for the VGLS module and the method for generating geographic coordinates from the final selected area need to be elaborated to enhance the methodological rigor.

**Other Comments Or Suggestions:**

Before the “Prompt Consideration”, it should be a full stop. The description for the next location prediction should be consistent as it is denoted as next-location prediction in the Introduction.

**Other Strengths And Weaknesses:**

Pros:
1.	The paper demonstrates strong originality by applying vision-language models (VLMs) to next-location prediction, a novel extension beyond conventional VLM tasks.
2.	The proposed Visual Guided Location Search (VGLS) introduces a creative and effective recursive refinement strategy that leverages visual reasoning for spatial prediction, demonstrating potential for training-free generalization across cities.
3.	The construction of a large-scale dataset combining real-world taxi trajectories from Porto and Chengdu, along with a well-designed human benchmark based on over 10,000 predictions, provides a robust and meaningful evaluation framework.
Cons:
1.	The recursive partitioning strategy in VGLS, while effective, may introduce high computational costs as the number of iterations increases, and the lack of a clear stopping criterion creates ambiguity in the refinement process. Moreover, how to obtain the estimated geographic coordinates after the area search remains to be clarified.
2.	The comparison with baseline models is somewhat limited, as the paper primarily uses a simple RNN and human performance without benchmarking against more sophisticated trajectory prediction models.
3.	The analysis of experiments is relatively shallow. It does not explore why Claude demonstrates superior performance. Investigating the architectural or training differences that give Claude an advantage could provide valuable insights for designing future vision-language models with enhanced spatial reasoning and general intelligence capabilities.
4.	The paper lacks substantial theoretical innovation, as the proposed recursive partitioning strategy (VGLS) primarily builds on existing hierarchical search and visual reasoning techniques without introducing fundamentally new theoretical insights. Additionally, the practical applicability of the method remains limited, as the current performance of VLMs still falls short of human-level accuracy in most scenarios, indicating that further improvements are needed before the approach can be reliably deployed in real-world applications.

**Questions For Authors:**

1. What are the underlying factors contributing to Claude 3.5 Sonet’s superior performance in next location prediction compared to other VLMs?
Understanding why Claude outperforms other models could provide valuable insights for improving spatial reasoning in VLMs. If the authors can identify specific architectural or training differences responsible for Claude’s advantage, it would strengthen the paper’s contribution to the design of future VLMs.
2. What insights can be drawn from the model's failure cases, and how could they inform future improvements?
A more detailed analysis of the failure patterns (e.g., trajectory complexity, road network alignment) could reveal structural limitations in the VGLS approach and suggest avenues for refining the model’s decision-making process. Moreover, the improvement of the prompt design is also an interesting issue.
3. Does the method of image partitioning (e.g., left-right splitting versus triangular splitting), the order of partitioning, image scaling and image orientation affect prediction performance?
Clarifying whether different splitting strategies or orders influence the accuracy and consistency of the model’s predictions would provide deeper insights into the robustness of the VGLS approach. Additionally, understanding whether image scaling and orientation adjustments affect model performance could help refine the preprocessing pipeline and improve generalization across different map formats and resolutions.

**Relation To Broader Scientific Literature:**

The paper makes meaningful contributions at the intersection of next-location prediction, vision-language modeling, and spatial reasoning. It extends VLMs beyond their traditional use cases, introducing a training-free approach that generalizes across different environments. More parameters of model results in stronger reasoning for the next location, which aligns with prior conclusion of the number of parameters is crucial for the ability of model inference.

**Theoretical Claims:**

The paper does not present any formal theoretical proofs or rigorous mathematical derivations. The proposed Visual Guided Location Search (VGLS) module is primarily described as a procedural algorithm rather than a theoretical framework supported by formal proofs. While the paper introduces a recursive refinement strategy and discusses its logical foundation, it does not attempt to formally prove the convergence or optimality of the VGLS process. Therefore, there are no theoretical claims that require validation through mathematical proofs. The authors could consider providing a formal analysis of the convergence properties, computational complexity, and potential error bounds of the recursive search process. This would enhance the theoretical robustness of the proposed method.

---

> ### Author Rebuttal · Authors · 2025-04-01
>
> Thank you for your comments. Below, we respond to your concerns. Due to space constraints, some points may be addressed briefly; please feel free to raise any questions.
>
> **On the Superior Performance of Claude Models**
>
> The Claude series consistently achieves SOTA results on spatial reasoning tasks [1]. Researchers speculate that Claude models may possess a well-developed world model, likely due to **optimization for screen control tasks**, which require fine-grained spatial understanding of interface layouts, highlighting its spatial reasoning capabilities.
>
> **On Novelty**
>
> The novelty of this work lies in introducing the **first benchmark for next-location prediction based on vision-language models**. Since road network structures vary across cities, training city-specific models limits generalization. In contrast, vision encoders provide **universal map representations**, enabling cross-city generalization. Promising future path include: Reinforcement learning with reward signals, Visual fine-tuning via LoRA. We leave these directions to future work and hope this benchmark inspires broader research in vision-based location prediction.
>
> **On the Formal Convergence of Our Method**
>
> Our method assumes that VLMs inherently possess an **internal capability** to predict the next location, but this capability may not be directly shown because of inability to paint on the image. We therefore reformulate the generative task as a discriminative one. For an image of size HxW, the distance between the model’s selected location and its internal prediction is bounded by $\sqrt {H^2+W^2}/{2^i}$ after i steps. This distance converges to 0 as the number of steps increases.
>
> **Insights from Failure Cases**
>
> We categorize the identifiable errors into two types: **Visual hallucinations**: In some cases, the region of interest is too small for the model to differentiate, potentially due to the patching mechanism in the vision encoder. **Speed-related biases**: As shown in Porto Case 41, an abnormally high-speed segment caused the model to overemphasize that portion, leading to an incorrect prediction. This may reflect a limitation in the model’s attention mechanism.
>
> **On the Influence of Other Trajectory Factors**
>
> In Appendix B.1, we analyze several additional factors: **Image scaling**. When a unit length represents a shorter real-world distance, performance improves. **Angular change**. Larger angular shifts tend to worsen accuracy. **Path length**. Longer trajectories correlate with higher errors. Experiments related to transparency, background color, and trajectory color will be included in the appendix if the paper is accepted.
>
> **Regarding the Statement on Insufficient Data for Training RNNs**
>
> A simple motivating example is the prediction of next-location trajectories for **elderly individuals**. GPS data for this demographic is extremely sparse due to the need for specialized data collection equipment. As a result, it is difficult to train RNN-based models effectively in such cases. Leveraging the generalization ability of VLM offers a way to bypass the need for domain-specific, large-scale trajectory data.
>
> **On the Transferability Limitations of Other Models**
>
> As mentioned in prior works, models such as LLM-Mob and Agent-Move require training a dedicated token for each region within a specific city. These tokens are tightly coupled with the city they are trained on, significantly limiting the transferability of these models across cities.
>
> **On Consistent Instructions for Human Participants**
>
> We have stated in Line 167 (right column) of our paper, *“Users are presented with the same input prompts”* This ensures fair comparison.
>
> **On the Choice of Distance Thresholds**
>
> **100m** roughly corresponds to the size of a football field. **500m** typically covers the space between traffic signals. **2000m** approximates the span of a city block.  Each threshold offers meaningful granularity for evaluating real-world spatial accuracy.
>
> **On the Number of Iteration Steps**
>
> As mentioned previously, increasing the number of iterations reduces prediction error, akin to inference-time scaling in large models. We chose to terminate after 10 rounds, at which point the average distance between the predicted region center and the model's internal prediction was **under 32 meters**, and we consider it acceptable. The final prediction is defined as **the center of the region** selected in the 10th round.
>
> **On Comparison with More Sophisticated Trajectory Models**
>
> Following your suggestion, we now include results for **four SOTA models**. Due to space constraints, we only report the average MAE across all datasets here: Transformer: 216.73, DeepMove: 215.18, GETNext: 214.33, LLM-Mob: 211.12. We will include full experimental results in the final version of the paper if accepted.
>
> Sincerely,
> The authors of Paper 740
>
> [1] https://mcbench.ai/leaderboard
>
> [2] https://www.anthropic.com/news/visible-extended-thinking

---

### Official Review · Reviewer_DCmP · 2025-03-23

**Overall Recommendation:** 1

**Summary:**

This paper introduces a new task, next-location prediction, which leverages map images and historical coordinates to predict the next location. The paper proposes a framework, VLMLocPredictor, which guides VLMs to iteratively refine the next-location prediction. Moreover, the paper compares the performance of multiple VLMs and humans on this task, provides discussions on the results, and offers further analyses of several influential factors.

**Claims And Evidence:**

Yes.

**Essential References Not Discussed:**

No.

**Experimental Designs Or Analyses:**

I am a bit skeptical of the validity of the experimental designs in this paper. To predict the next location, the input information seems to lack sufficient conditions.

**Methods And Evaluation Criteria:**

I have questions regarding the benchmark datasets used in this paper. Merely providing images of road networks and historical trajectories seems insufficient to predict the next location on a map scale.

**Other Comments Or Suggestions:**

Please see the weaknesses.

**Other Strengths And Weaknesses:**

Strengths:

1. This paper is comprehensive in content. From introducing a new task, providing corresponding solutions, benchmarking the performance of multiple VLMs and humans, to final discussions and analyses, it covers a wide range of aspects.

2. The paper conducts extensive experiments and provides various discussions. The analyses include the performance of predicting locations on the map by VLMs and humans, as well as several influential factors, such as prompts.

Weaknesses:

1. There may be issues with the experimental design. Next locations on the navigation map are highly dependent on the intention or destination of the drivers and are influenced by factors such as dynamic traffic conditions. Merely providing images of the road network and historical trajectories seems insufficient to predict the next location. Providing insufficient information may lead to guesses based on inadequate conditions.

2. The analysis provided in this paper, such as the color choices, appears to be somewhat superficial. The results of ablation studies show that many attempts result in only minor differences. Given sufficient information, it would be better to introduce research challenges with scientific value and provide corresponding solutions, rather than a simple enumeration of conditions.

**Questions For Authors:**

Could you provide an in-depth summary of the scientific insights for the research community or the application values for real-world scenarios in the paper, to illustrate its broader influence and significance?

**Relation To Broader Scientific Literature:**

The contributions are related to the scientific literature on the application of VLMs to map image processing.

**Theoretical Claims:**

The major claims in this paper are theoretically correct.

---

> ### Author Rebuttal · Authors · 2025-03-31
>
> Thank you for your valuable comments. We appreciate the time and effort you have devoted to reviewing our work. Below, we respond to your concerns in detail.
>
> ### **Regarding Experimental Design**
>
> 1. While it is true that taxi drivers' trajectories are influenced by intent, prior work suggests that human mobility remains highly predictable despite this. Notably, the Science paper *"Limits of Predictability in Human Mobility"* (2010), which has been cited over 4,000 times [1], showed that for hourly human mobility data, the theoretical upper bound of next-location prediction accuracy can reach as high as **93%**. This indicates that, although behavior is goal-driven, there are still strong patterns and structural constraints that make the task meaningful and predictable.
> 2. In addition, the topological structure of road networks imposes natural constraints on plausible movements. Consider a simple example: suppose the vertices of a square are labeled A, B, C, and D in order. If a driver travels from A to B to C, and all roads are open and unconstrained, it is unlikely that the next step will be D—since a more efficient plan would have been a direct route from A to D. This highlights that predicting the next location depends on a model’s ability to reason about spatial efficiency and road network topology.
> 3. Lastly, as larger models consistently achieve better results,  which is consistent with scaling laws in deep learning, and no model has yet reached human-level performance, it demonstrates that the behavior of taxi driver can be in-deed inferred.
>
> For these reasons, we respectfully ask the reviewer to reconsider their concerns about the experimental design.
>
> ### Regarding Research Challenges
>
> 1. We agree that experiments like *color choice* are relatively simple, which is why they were included only in the appendix. However, they serve to illustrate that our benchmark design is deliberate and well-grounded. Moreover, in our ablation studies, we show that carefully designed prompts reduce model error by over 22%, indicating a meaningful impact.
> 2. As for the scientific value of the research challenge: as mentioned earlier, our benchmark directly targets the reasoning capabilities of VLMs. Road networks are complex topological graphs, and predicting trajectory continuations requires non-trivial structural reasoning. Our benchmark provides a first step in evaluating whether models can perform human-like spatial reasoning over such structures.
>
> While we do not focus on improving model accuracy in this work, we see this as a **rich direction for future research**, and suggest a few promising directions:
>
> - **Reinforcement learning with reward signals**: Since selecting the correct region (e.g., color) is straightforward to verify, one could define reward functions and train models to improve their reasoning through interaction.
> - **Visual fine-tuning with LoRA**: For example, training a vision-language generation model to take a map as input and output a predicted next-point image. These approaches could significantly improve VLM performance on our benchmark, and we leave their exploration to future work.
>
> We hope this clarifies the scientific challenges and potential of the benchmark.
>
> ### **Regarding In-Depth Summary of the Scientific Insights**
>
> 1. **Vision-based Next-Location Prediction**:
>    Because the visual representation of road networks is domain-agnostic, VLMs offer the potential to build generalizable next-location predictors. Our benchmark establishes a unified framework for evaluating such models, not limited to VLMs alone.
> 2. **Reasoning over Complex Road Networks**:
>    As emphasized, our benchmark is designed to test whether VLMs can reason over complex topological structures, which is central to understanding spatial intelligence and planning. From the experimental results, the large models exhibit the complex reasoning ability to some extent.
>
> Once again, thank you for your thoughtful feedback and the time you’ve dedicated to our work. If you have any further questions or if any concerns remain unaddressed, please feel free to reach out—we would be happy to continue the discussion.
>
> Sincerely,
> The authors of Paper 740
>
> [1] Song, Chaoming, et al. "Limits of predictability in human mobility." *Science* 327.5968 (2010): 1018-1021.

---

### Decision · Program_Chairs · 2025-05-01

**Decision:**

Reject

**Comment:**

After the discussion phase, the majority of reviewers leaned towards rejection (Weak Accept, Weak Reject, Reject). R-Lpbt, who provided the most thorough review, acknowledged the novelty of applying VLMs to the next-location prediction task and recognized the potential impact of the introduced dataset. However, they raised several concerns, including missing technical details (e.g., unclear stopping criterion in VGLS), high computational costs, limited experimental analysis with few baseline comparisons, lack of substantial theoretical innovation, and limited practical applicability. The rebuttal addressed some of these concerns and introduced new baseline comparisons, but R-Lpbt noted that their concerns remained mostly unresolved. Additionally, the AC agrees with R-Lpbt that the experiments do not support the claim that the proposed method is more robust to cross-city generalization, which is the primary motivation behind the design of the method. As a result, the AC believes the manuscript would benefit from further revisions and a decision to reject was reached.